# Molecular Mechanisms and Clinical Challenges of Glioma Invasion

**DOI:** 10.3390/brainsci12020291

**Published:** 2022-02-20

**Authors:** Tomoya Oishi, Shinichiro Koizumi, Kazuhiko Kurozumi

**Affiliations:** Department of Neurosurgery, Hamamatsu University School of Medicine, Hamamatsu 431-3192, Japan; coizmmd@hama-med.ac.jp (S.K.); kurozu20@hama-med.ac.jp (K.K.)

**Keywords:** glioma, invasion, extracellular matrix, glioma therapy-related invasion

## Abstract

Glioma is the most common primary brain tumor, and its prognosis is poor. Glioma cells are highly invasive to the brain parenchyma. It is difficult to achieve complete resection due to the nature of the brain tissue, and tumors that invade the parenchyma often recur. The invasiveness of tumor cells has been studied from various aspects, and the related molecular mechanisms are gradually becoming clear. Cell adhesion factors and extracellular matrix factors have a strong influence on glioma invasion. The molecular mechanisms that enhance the invasiveness of glioma stem cells, which have been investigated in recent years, have also been clarified. In addition, it has been discussed from both basic and clinical perspectives that current therapies can alter the invasiveness of tumors, and there is a need to develop therapeutic approaches to glioma invasion in the future. In this review, we will summarize the factors that influence the invasiveness of glioma based on the environment of tumor cells and tissues, and describe the impact of the treatment of glioma on invasion in terms of molecular biology, and the novel therapies for invasion that are currently being developed.

## 1. Introduction

Gliomas are primary brain tumors that arise in the brain parenchyma and have histologically similar features to normal glial cells. Of these, glioblastoma is the most common tumor in adults and is a biologically aggressive tumor characterized by high cell density, pleomorphic tumors with mitosis, and either microvascular proliferation or necrosis [1]. The extent of resection is the most important independent predictor of overall survival (OS) and progression-free survival (PFS) [2], and an extent of resection of 78% or higher is required to improve prognosis [3]. Although the development of assistive technologies, such as awake surgery, intraoperative navigation, intraoperative magnetic resonance imaging, and 5-amino levulinic acid (5-ALA), has improved the removal rate [4,5], the prognosis for standard treatment remains unsatisfactory at approximately 15 months [2]. Tumor cells are highly infiltrative and often invade important brain regions, making it very difficult to obtain a negative tumor margin. Even if a total resection is achieved, patients often suffer from recurrence around the extraction cavity. Therefore, the control of invasive lesions is one of the issues to be solved in the treatment of glioma. This article will review the molecular mechanisms of glioma invasion, the impact of glioma treatment on invasion, and the developing treatments for glioma invasion.

## 2. The Characteristics of Glioma Invasion

### 2.1. Cell Dynamics

Unlike metastatic brain tumors, glial-derived tumors are prone to invade the normal brain. “Cell migration” is defined as the movement of cells from their original location to another, whereas “cell invasion” is defined as the ability of cells to navigate through the extracellular matrix within a tissue or to infiltrate neighboring tissues [6]. Tumor cell invasion involves four steps: (1) detachment from the primary tumor mass, (2) adhesion to the extracellular matrix (ECM), (3) degradation of the ECM, and (4) movement and stretching of the invading cells [7,8]. The invasion of glioma follows the nature of neural progenitor cells [9], while glioma cells take the form of individual cells and cell clusters [7,8,10] and infiltrate along blood vessels and nerve fibers by saltatory migration [11,12,13]. The migration speed has been measured to be 51 to 100 μm/h [13,14,15]. Cell division is often observed at the bifurcation of blood vessels [14].

### 2.2. Invasion to the Corpus Callosum and Subventricular Zone

The infiltration of glioma often presents with a butterfly appearance. Invasion through the corpus callosum is seen in 14% of cases, and gliomas presenting with invasion of the corpus callosum are more aggressive [16,17]. In highly invasive gliomas, invasion of the subventricular zone is also often seen. In particular, glioma stem cells (GSCs) are prone to invade the subventricular zone [18]. Invasion of the subventricular zone has a poor prognosis and high recurrence rate [19,20,21]. Involvement of the subventricular zone is associated with a high expression of pleiotrophin (PTN), also known as a heparin-binding growth-associated molecule, which is exerted by neural progenitor cells. PTN binds to secreted protein acidic and rich in cysteine (SPARC)/SPARC-like protein 1 (SPARCL1) and heat shock protein 90B (HSP90B) as the partners, activates Rho/Rho-associated protein kinase (ROCK) signaling, and promotes cell migration [22]. In addition, there are two types of PTNs: immobilized pleiotrophin (PTN18), which promotes migration via the cell surface receptor, the protein tyrosine phosphatase receptor zeta (PTPRZ1), and soluble pleiotrophin (PTN15), which is mainly involved in promoting glioblastoma proliferation [23]. PTN is a strong binder of glycosaminoglycans (GAGs) and has been shown to interact with a variety of receptors, including proteoglycans PTPRZ and syndecans and GAG non-containing integrin and nucleolin [24]. These interactions depend on the sulfation density of GAGs and activate many intracellular kinases, which are involved in cell activation and transformation [24,25].

Glioma infiltration of the subventricular zone is also related to C-X-C motif chemokine receptor type 4 (CXCR4)/C-X-C motif chemokine ligand 12 (CXCL12 or stromal derived factor-1, SDF-1) [26]. The CXCR4/CXCL12 axis upregulates the downstream phosphoinositide-3 kinase/serine-threonine protein kinase B/nuclear factor-kappa B (PI3K/Akt/NF-κB) pathway and is involved in cell survival, migration, and stemness [27]. Compared to non-invasive tumor cells, gliomas have higher expression of CXCR4 [28]. CXCL12 induces the invadopodia formation and the expression of membrane type-2 matrix metalloproteinase (MT2-MMP), which degrades the surrounding ECM and is involved in glioma invasion [29,30]. Invadopodia consist of actin-rich protrusions that facilitate the invasion of tumor cells from the tumor cell mass to the surrounding healthy parenchyma [31]. MMPs are enriched and secreted at the tips of invadopodia, mediating the degradation of the ECM [32].

## 3. Hypoxia

One of the mechanisms of cancer cell invasion is hypoxia-driven motility, which is enhanced in hypoxic conditions [33]. Although oxygen is essential for the maintenance of cell life, cancer cells proliferate so rapidly that insufficient angiogenesis results in the formation of hypoxic areas. Hypoxia-inducible factor (HIF)-1α is an important factor in hypoxic conditions. Semenza et al. found that HIF-1α is upregulated in cancer cells under hypoxic conditions [34]. HIF-1α is normally subjected to prolyl hydroxylation under normal oxygen conditions, and is further degraded by proteosomes after ubiquitination by von Hippel–Lindau. In contrast, in hypoxic conditions, HIF-1α is not hydroxylated and is transferred to the nucleus. As a result, it binds to HIF-1β and causes various gene expressions related to angiogenesis, migration, cell survival, and glucose metabolism [35].

Activation of the HIF-1 pathway is a common feature in glioma, and HIF-1 regulates target genes in activators of angiogenesis and invasion in glioma [36]. In hypoxic areas, HIF accumulates and enhances glioblastoma invasiveness through increased delta like non-canonical Notch ligand 1 (DLK1) expression [37]. HIF-1α stabilizes and upregulates the Notch intracellular domain (NICD) to activate the Notch pathway, which is involved in maintaining GSCs [38] and enhancing cell invasion [39]. Activation of the PI3K/Akt/mTOR pathway by HIF-1α has also been reported to cause enhanced invasiveness [40]. Thus, the HIF-1 pathway contributes significantly to the invasiveness of glioma.

## 4. Factors Associated with ECM

Tumor cell adhesion and degradation to the ECM, which is important for tumor cell invasion, involves various factors at the cell surface and in the intercellular space. The tumor microenvironment provides gliomas with invasion, proliferation, and resistance to treatment. The extracellular matrix plays roles in scaffolding and the maintenance of tissue homeostasis, and its degradation and changes have a major impact on cancer invasion. Factors involved in ECM adhesion include integrin, brain-specific angiogenesis inhibitor (BAI1), cysteine-rich 61/connective tissue growth factor/nephroblastoma overexpressed (CCN1), proteoglycans, fibronectin, laminin, cadherin, collagen, CD44, and factors involved in degradation include protease, such as matrix metalloproteinases (MMPs), a tissue inhibitor of MMP (TIMP) [8]. In this section, we describe MMPs, integrin, CCN1, and proteoglycans. The ECM and cell surface factors involved in glioma invasion are summarized in Figure 1.

### 4.1. Matrix Metalloproteinases (MMPs)

MMPs are members of the zinc-dependent endoproteases family that play a role in ECM remodeling by degrading the proteins responsible for various ECM structures. In malignant tumors, MMPs promote invasion and metastasis behavior in the epithelial-mesenchymal transition (EMT) [41]. MMP-2 and MMP-9 are the most well-studied and major promoters of tumor cell invasion. The expression of MMP-2 in glioma has been reported to be an important key molecule involved in malignancy and invasion [42,43]. MMPs are promoted by transforming growth factor (TGF)-β, a key molecule in the EMT, while TIMPs are suppressed by TGF-β [41,44,45]. In addition, tumor cells use aerobic glycolysis in energy metabolism despite adequate oxygenation; excess extracellular lactate enhances the expression of MMP-2 and integrin αvβ3 via high expression of TGF-β2, which enhances glioma cell migration [46,47]. Heparanase (HPSE) degrades heparan sulfate and shortens the heparan sulfate chains on Syndecan-1, which makes the core protein susceptible to degradation by protease. Furthermore, HPSE is involved in tumor metastasis and angiogenesis by regulating the expression of downstream effector genes such as HGF, MMP-9, and VEGF [48].

### 4.2. Integrin

Integrins are cell surface proteins that are key molecules involved in cell-extracellular matrix adhesion and cell-cell adhesion, and play a role in initiating various signaling cascades through the binding of α and β subunits. Twenty-four heterodimers are formed from 18 α-subunits and 8 β-subunits, and the ligand preference is determined by collagen-, laminin-, RGD motif-binding-, and leucocyte-specific receptors. Binding of integrins to the ECM promotes proliferation, invasion, and metastasis. Integrins enhance pathways such as the PI3K/AKT pathway, RAS, or small GTPases and mitogen-activated protein kinase (MAPK). In glioblastomas, αvβ3 and αvβ5 are upregulated, and αvβ3 co-localizes with MMP-2 in tumor cells [49]. TGF-β promotes glioma cell migration via αvβ3 integrin expression [50]. Collagen accumulation and crosslinking increase ECM stiffness and integrin clustering promotes focal adhesions and drives tumor invasion [51,52].

### 4.3. Cysteine-Rich 61/Connective Tissue Growth Factor/Nephroblastoma Overexpressed (CCN1)

The Cysteine-rich 61/connective tissue growth factor/nephroblastoma overexpressed (CCN) protein family is found on the ECM and cell surface and is involved in cell-matrix interactions, such as cell proliferation, attachment, migration, differentiation, wound healing, and angiogenesis. CCN1 interacts with α6β1, αvβ3, αvβ5, and αIIβ3 integrins to trigger downstream signals such as PI3K/Akt, TGF-β, and vascular endothelial growth factor (VEGF) signaling [53,54,55]. CCN1 has been reported to be overexpressed in 48 to 69% of primary gliomas and is associated with PFS and OS [55,56]. CCN1 is secreted by differentiated glioblastoma cells rather than glioma stem cells, which promotes the migration of macrophages into the tumor and contributes to GSC-dependent tumor progression [57].

### 4.4. Proteoglycans

The major extracellular matrix components of the adult brain are glycosaminoglycan hyaluronic acid, proteoglycans of the lectican family, and link proteins [44]. Proteoglycans (PGs) are molecules consisting of a core protein and GAG side chains such as chondroitin sulfate (CS) and heparan sulfate (HS). Chondroitin sulfate proteoglycans (CSPGs) are critical regulators of brain tumor histopathology, and the low content of CSPGs is related to the active invasion of glioma cells [58]. Extracellular proteoglycans can bind to matrix proteins, trapping ligands such as growth factors [59].

The lectical subfamily of CSPG includes aggrecan, syndecan, neurocan, versican, and brevican. Syndecan-1, a transmembrane HS proteoglycan, is particularly upregulated in glioblastoma and is activated in an NFκB-dependent manner. Syndecan-1 interacts with HPSE and enhances growth factor signaling to promote the growth of glioma cells [60,61,62]. Moreover, brevican, a member of the lectican family of CSPG is upregulated in glio mas and its expression induces glioma invasion, which is especially enriched in the glioma stem cell (GSC) niche [63]. The HPSE is an endo-β-D-glucuronidase that degrades the heparan sulfate side chain of HSPG. It is an important regulator of ECM remodeling and is involved in the growth and invasion of glioma [64]. HPSE upregulates extracellular signal-regulated kinase (ERK) and AKT pathways to increase glioma cell proliferation and worsen prognosis [64,65].

## 5. Glioma Stem Cells (GCS)

Glioma stem cells (GSCs), also called glioma-initiating cells, are cellular subpopulations that, like normal stem cells, are capable of self-renewal and differentiation to produce secondary tumors. A heterogeneity is created by cells that have a tendency to differentiate from the GSC at the top of the hierarchy [66]. Among them, GSCs are responsible for the distinctive feature of glioma invasion [67]. GSCs are characterized by the presence of CD133, CD44, leucine-rich repeat-containing G protein-coupled receptor 5 (LGR5) as surface markers [68]. The factors involved in the invasion of glioma stem cells are summarized in Figure 2.

The expression of delta-like canonical Notch ligand (DLL)-1, Notch1, nestin, and Sox2 is upregulated in GSCs compared to conventional cell lines [69]. In particular, Notch is a signal that functions in cell fate determination during tissue construction, and is involved in the maintenance and regulation of neural stem cells and progenitor cells differentiation during central nervous system development [70]. In white-matter GSCs, Notch-induced transcription factor Sox9 upregulates Sox2 and attenuates Notch1 promoter methylation to enhance Notch1 expression. This positive feedback loop increases invasiveness and worsens the prognosis [71]. Notch1 also activates the PI3K/Akt pathway and stimulates β-catenin and NF-κB signaling to promote the migratory and invasive properties of glioma [69,72,73]. In addition, Notch1 signaling upregulates CXCR4 expression and activates the CXCL12/CXCR4 autocrine/paracrine loop to enhance GSC survival and invasiveness [69,74]. Thus, signaling through increased Notch expression in glioblastoma, especially in GSCs, is thought to be highly relevant to tumor invasion, and Notch inhibitors are being developed [70,75].

CD44 is a transmembrane glycoprotein that mediates cell-cell or cell-matrix interactions with hyaluronic acid (HA) as the main ligand, and is strongly involved in tumor progression, apoptosis evasion, multidrug resistance, and cell invasion. CD44-mediated signaling has been implicated in MMP-mediated matrix degradation, tumor growth, and tumor invasion [76]. HA is a major component of white matter, a frequent route of glioma invasion, and increases the invasiveness of glioblastoma in a dose-dependent manner [77,78]. CD44 binding with HA stimulates a number of signaling pathways, such as PI3K/Akt/mTOR, Ras, focal adhesion kinase (FAK), and ERK signaling, and induces MMP-9 expression [79,80]. Receptors for HA-mediated cell motility (RHAMM), CD44, and osteopontin, which is a ligand for CD44, are involved in HA-mediated migration, invasion, proliferation, radiation therapy, and chemotherapy resistance [80,81]. Tumor-associated mesenchymal stem cells interact with glioblastoma and increase invasiveness by remodeling the ECM. Tumor-associated mesenchymal stem cells increase HA levels in the ECM by upregulating HA synthase-2 (HAS2) expression [82].

## 6. Epithelial-Mesenchymal Transition (EMT)

Cell invasion requires a reduction in the adhesive connections that maintain cell-to-cell adhesion. The EMT is a reversible change found in cells of epithelial origin, and mesenchymal phenotype changes are associated with increased cell motility and resistance to apoptosis [83]. In general, the EMT produces cell detachment from the basement membrane and the formation of a mass of mesenchymal cells at sites away from the origin. EMTs are classified into three subtypes according to the biological setting. Type 1 EMT is associated with embryogenesis and gives rise to the mesoderm and endoderm and to mobile neural crest cells. Type 2 EMT is a program that begins as part of a tissue repairment that normally generates fibroblasts and other related cells in order to reconstruct tissues following trauma and inflammatory injury. Unlike these subtypes, type 3 EMT occurs in neoplastic cells. Carcinoma cells undergoing a type 3 EMT may invade and metastasize [83]. The term proneural–mesenchymal transition is also used for glioma, as well as EMTs for other aggressive cancers [84]. The EMT is an important driver of invasiveness and recurrence of glioblastoma, with cellular reprogramming causing cytoskeletal remodeling and loss of adhesion molecules [85]. The main executors of the EMT are EMT-activated transcription factors (EMT-TF), such as SNAIL, TWIST, and ZEB family [86]. SNAIL induces MMP-9 expression triggered by TGF-β [87]. Notch signaling is required for the conversion of hypoxic stimuli into the EMT [39].

TGF-β is a major key molecule that induces the EMT via various transcription factors [88]. TGF-β is an important cytokine that maintains homeostasis, and the TGF-β pathway acts as an oncogenic factor to induce angiogenesis, immunosuppression, cell invasion, and proliferation in tumor progression, including glioblastoma [89,90]. TGF-β1 activates a variety of downstream signaling pathways, including PI3K, Smads, and MAPK, which are key players of the TGF-β-induced EMT [83,86]. Proteolytic degradation by MMPs plays a central role in the EMT process, and the EMT-related pathway is one of the regulatory mechanisms for MMP expression. In oral squamous cell carcinoma, TGF-β1 facilitates MT1-MMP-mediated MMP-9 activation and stimulates invasion of the tumor in collaboration with MT1-MMP and MMP-2 [91]. Elevated TGF-β activity is associated with poor prognosis in glioma patients [89,92]. In addition, TGF-β is also involved in tumor initiation and recurrence via CD44 and inhibitors of DNA-binding protein (Id)-1 [93].

## 7. Effect of Glioma Therapy on Tumor Invasion

As mentioned above, various molecular signaling pathways are intricately involved in the invasion of glioma cells. In the current treatment of glioma, the preclinical impact on glioma invasion is being studied.

### 7.1. Radiation Therapy

The standard of care for glioma is postoperative chemotherapy and radiation therapy. Radiation therapy is the main treatment modality for glioma lesions that cannot be safely resected. Whether or not the post-radiation microenvironment enhances invasiveness is inconclusive. It has been reported that radiotherapy coupled with temozolomide (TMZ) treatment has an additional effect of inhibiting the proliferation and migration of glioma spheroids [94]. It is thought that the radiation-induced tumor bed effect reduces blood flow, pH, and hypoxia, making the environment unsuitable for tumor cell survival [95,96]. However, it has been pointed out that radiation increases the invasiveness and motility of glioma [97,98]. Tsuji et al. reported that the secretion of CXCL12, VEGF-A, TGF-β1, and TNFα is enhanced in the brain after irradiation, and that the microenvironment in the brain in the chronic phase after irradiation is suitable for tumor cell growth and invasion [99]. In experiments using cell lines, various changes have been observed in invasiveness caused by radiation [100,101]. Radiation-induced damage of the tumor microenvironment may create a tumor-susceptive niche that promotes the proliferation and invasion of the residual glioma cells [102]. In a model experiment, multiple radiations altered glycosylated components (PG and GAG) in normal brain tissue, reduced CSPG expression and CS in normal brain tissue, and promoted residual glioma cell adhesion and proliferation [103]. In molecular biology, radiation has been reported to activate MMP-2 and MMP-9 through p53, resulting in increased invasiveness [104,105]. It was also suggested that radiation increases the activity of MMP-2 and MMP-9 through the expression of integrin αvβ3 [106]. PI3K-mediated activation of the Rho signaling pathway is associated with radiation-induced invasion [98]. HIF-1α is also an important molecule that contributes to radiation-induced enhancement of invasiveness. It has been reported that irradiation stabilizes HIF-1α by destabilizing prolyl hydroxylases (PHD)-2 and protein von Hippel-Lindau (pVHL) [107]. Ionizing radiation enhances the invasive capacity of GSCs through stabilization of HIF-1α and activation of junction-mediated protein [108].

However, although radiation therapy may change the tumor microenvironment to increase the invasiveness of tumors, radiation is still an important modality for the treatment of tumors. It has been reported that high linear energy transfer (LET) irradiation, such as alpha and carbon radiation, suppresses migration, and future development of radiotherapy is expected [109,110].

### 7.2. Temozolomide (TMZ)

TMZ is an alkylating oral anticancer drug that has been used in conjunction with postoperative radiation therapy as the standard treatment for glioma [2]. It is one of few anticancer drugs that can pass through the blood-brain barrier (BBB) and is the mainstay of postoperative chemoradiotherapy in the current treatment of glioma. It is unclear whether TMZ enhances the invasion of glioma. MMP-2 secretion and invadopodia formation is enhanced by radiation and TMZ therapy [111,112]. There are reports that the expression of CXCR4 and VEGF and the activity of MMP-2 and MMP-9 were reduced by TMZ [113,114].

### 7.3. Anti-VEGF Therapy

VEGF is a stimulator of angiogenesis that is frequently expressed in glioblastoma; it is commonly attributed to the autocrine and paracrine production of VEGF-A. Inhibiting VEGF signaling suppresses the tumor growth of glioma xenografts in model mice [115,116]. Anti-VEGF antibody is a monoclonal antibody to VEGF and has a certain effect on the tumor control of primary or relapsed glioblastoma [117,118,119]. Although it improved PFS in primary and recurrent glioblastoma, it was not effective in improving OS.

There is some preclinical evidence that antiangiogenic therapies promote glioma cell invasiveness. Anti-VEGF therapy induces a vascular gradient, which, in turn, induces tumor hypoxia, macrophage infiltration, mesenchymal transition, stem cell marker expression, and increased invasiveness [120]. It has been reported that hypoxia induced by anti-VEGF enhances angiogenesis, tumor survival, invasion, and resistance to therapy. Keunen et al. reported that, in a rat-patient-derived xenograft model, bevacizumab treatment resulted in a decrease in tumor volume and tumor blood flow, but a 68% increase in infiltrating cells, which was associated with the enhanced expression of HIF-1 and activation of the PI3K pathway and Wnt-signaling pathway [121]. Shimizu et al. also reported that bevacizumab upregulates δ-catenin in glioma cells and increases invasiveness [116]. It has been reported that suppression of VEGF increased CD44 expression and that GSCs became invasive [122]. Administration of bevacizumab causes the dose-dependent accumulation of collagen, MMP-2, and MMP-9, which play important roles in the adhesion process of tumor cell invasion and degradation of the cellular matrix [123]. Although invasiveness is perhaps enhanced by anti-VEGF therapy, the prognosis is not necessarily poor [118,119,124]. Combination with drugs that suppress invasion, such as γ-secretase inhibitor, has been used in an attempt to mitigate the bevacizumab-induced invasive effect [125].

### 7.4. Glucocorticoid

Glioma is often associated with prominent cerebral edema, which can cause mass effect and elevated intracranial pressure, affecting the prognosis [126]. Glioblastoma-induced cerebral edema has been conventionally treated with dexamethasone (DEX). Glioblastoma-induced brain edema is associated with vasogenic edema due to extravasation by disruption of the BBB, and DEX improves edema by increasing the expression of tight junction genes that regulate the endothelial permeability of the BBB [127]. The effect of DEX on the invasive properties of glioma is not well understood. Luedi et al. reported that DEX increases invasion, proliferation, and angiogenesis in GSCs, and that patients with a high DEX-regulated gene signature derived from DEX-treated GSCs showed worse prognosis [128]. It has also been reported that glucocorticoid receptor-β interacts with β-catenin and is involved in proliferation and migration [129]. DEX also affects the tumor microenvironment. The combination of TMZ and DEX affects proteoglycan structure and composition in normal brain tissue, resulting in worsened brain ECM, which is favorable for the progression of residual glioma cells; a high DEX dose results in downregulation of the transcription of PG-coding genes, whereas a high DEX dose and TMZ predominantly affects the polysaccharide GAG chains of the molecules [130]. In contrast, DEX inhibits migration via the suppression of glucocorticoid receptor-dependent ERK1/2 MAPK pathway and MMP-2 secretion [131,132]. Guan et al. reported that DEX inhibits cell proliferation and promotes migration and invasion by upregulating aquaporin-1 (AQP1) in C6 cells [133].

## 8. Treatment of Invasive Glioma and Its Future Development

The suppression of glioma invasion is one of the therapeutic approaches in gliomas that spread invasively to the brain and offer limited resection. However, treatments that inhibit invasion are still being studied at present. In this section, we will discuss treatments for invasion that are undergoing clinical trials and therapeutic development.

### 8.1. Tumor-Treating Fields (TTF)

Tumor-treating fields (TTF) is a non-invasive treatment that uses electrode pads on the scalp to deliver a weak, sustained 100 to 300 kHz mid-frequency current to brain tumors. TTF works by selectively inhibiting the mitosis of brain tumor cells and can prolong survival by 4.9 months [134]. In addition to inhibiting cell proliferation, TTF has also been reported to inhibit EMT, endothelial cell angiogenesis, and migration by downregulating PI3K/Akt/NF-κB pathways [135].

### 8.2. Molecular Target Drugs

As our knowledge of the invasion of various cancers increases, therapies that target the molecular mechanisms that lead to cancer invasion are being developed. The major therapeutic agents that are currently in development are summarized in Table 1.

**MMP inhibitor**: Degradation of the ECM has been well observed in tumor tissues, and MMPs have been considered a good target for tumor invasion. However, the selectivity, low bioavailability, and low metabolic profile of broad-spectrum MMP inhibitors limited the efficacy of MMP inhibitors and did not justify continuation of the clinical trial [136]. MMP inhibitors were expected to have an effect on the invasiveness of tumor cells, but clinical trials for gliomas to date have not shown positive results. In a mouse model of colorectal carcinoma, AB0041 and AB0046, which are monoclonal anti-MMP-9 antibodies, were found to inhibit tumor growth and metastasis [137]. Since bevacizumab increases the expression of MMP-9 [138], anti-MMP-9 therapy may be effective against bevacizumab-induced resistance and invasiveness. Clinical trials of GS5745, a monoclonal antibody of MMP-9, are in progress.

**Integrin inhibitor**: Cilengitide was one of the earliest integrin antagonists to enter clinical trials [139]. In a rat model, bevacizumab treatment caused tumor cells to invade the brain parenchyma along with blood vessels, but cilengitide administration suppressed the invasion of tumor borders, suggesting the involvement of integrin in bevacizumab-induced glioma invasion [140]. The phase 2 study of cilengitide showed prolongation of median OS by adding cilengitide to standard treatment, and development was expected [141]; however, phase 3 studies such as CENTRIC failed to show OS prolongation [142] and development was discontinued. The addition of cilengitide to oncolytic virus therapy has been shown to enhance antitumor effects, and cilengitide may undergo re-evaluation as virus therapy develops [143].

**Notch inhibitor**: The expression of Notch is increased in GSCs, and Notch inhibition by γ-secretase inhibitor has been attempted. γ-secretase cleaves Notch to form the NICD, which translocates to the nucleus. γ-secretase inhibitors can inhibit Notch signaling. Jin et al. showed that MRK-003, an inhibitor of Notch and Akt phosphorylation, suppresses invasion but not mitosis when combined with MK-2206, an Akt phosphorylation inhibitor [144]. The Notch pathway blockade by γ-secretase inhibitor inhibited tumor growth and neurosphere formation in culture, and also prolonged survival in xenograft mice [145]. However, in a phase 2 study, RO4929097, a γ-secretase inhibitor, was evaluated for 6 month PFS and neurosphere formation in patients with recurrent glioblastoma, but failed to demonstrate efficacy [146]. Although no effective treatment for GSCs has been established yet, their treatment may remain an attractive strategy.

**TGF-β inhibitor**: TGF-β inhibitor has been reported to have inhibitory effects on the metastasis of breast, colon, pancreatic, and gastric cancers [147,148,149,150]. LY2109761, a TGF-β receptor inhibitor, not only enhanced the effects of radiotherapy but also inhibited cell migration via the SMAD4 signaling pathway [150]. In glioblastoma treatment, Zhang et al. reported that LY2109761 suppressed migration in vitro, and LY2109761 added to RT + TMZ significantly inhibited tumor growth in model mice [151]. Since it is not an invasive model, however, the inhibition of invasion in vivo was inconclusive. A phase 2b study of AP-12009 (trabedersen), a phosphorothioate antisense oligodeoxynucleotide, a specific target for the mRNA of TGFβ2, showed safety against chemotherapy in recurrent high-grade glioma and significant improvement in the 14-month tumor control rate in anaplastic astrocytoma. However, the phase 3 study, SAPPHIRE, was discontinued due to a lack of patients [152]. Compared with chemoradiotherapy, galunisertib, a TGF-β receptor inhibitor, had a higher disease control rate but shorter PFS and no difference in efficacy [153].

**PI3K inhibitor**: Tumor metastasis and invasion are enhanced by activating the PI3K/Akt pathway through regulating the expression of MMP [154]. In glioblastoma, the PI3K pathway is activated by changes in epidermal growth factor receptor (EGFR) amplification and phosphatase and tensin homolog (PTEN) mutation; and PTEN changes are a poor prognostic factor in glioblastoma [155]. It has been reported that a number of inhibitory agents of PI3K were preclinically effective in inhibiting cell proliferation and invasion [156,157,158,159]. The PI3K inhibitor PX-866 was well tolerated in a phase 1 trial, but failed to meet its predefined efficacy endpoint in a phase 2 trial in patients with recurrent glioblastoma [160]. A phase 1 study of voxtalisib, the PI3K/mTOR inhibitor, plus TMZ with or without radiotherapy, in patients with high-grade gliomas demonstrated favorable safety; however, no conclusion could be drawn regarding the efficacy because of the small number of patients and short follow-up [161]. Buparlisib, on oral pan-PI3K inhibitor, has high penetration across the BBB. A phase 2 study of buparlisib in recurrent glioblastoma patients showed minimal efficacy. The lack of tumor response was explained by incomplete PI3K inhibition in the tumor tissue [162].
brainsci-12-00291-t001_Table 1Table 1Molecular-targeted drugs and clinical trials related to glioma invasion.Inhibitor
Target MoleculeClinical Trial PhaseReferenceMMP inhibitorAG3340 (prinomastat)MMPPhase 2NCT00004200
GS5745 (andecaliximab)MMP-9Phase 1, ongoingNCT03631836Integrin inhibitorcilengitideintegrinPhase 3Stupp et al. (2014) [142]Notch inhibitorRO4929097γ-secretase inhibitorPhase 2Peereboom et al. (2021) [146]TGF-β inhibitorAP-12009 (trabedersen)TGF-β2Phase 3Bogdahn et al. (2011) [152]
LY2157299 (galunisertib)TGF-β receptor IPhase 1b/2aWick et al. (2020) [153]PI3K inhibitorPX-866PI3KPhase 2Pitz et al. (2015) [160]
XL-765, SAR245409 (voxtalisib)PI3K/mTORPhase 1Wen et al. (2015) [161]
NVP-BKM120 (buparlisib) PI3KPhase 2Wen et al. (2019) [162]

### 8.3. Stem Cell Therapy

In recent years, research has been conducted on therapies using stem cells for drug delivery. Neural stem cells, mesenchymal stem cells, and induced pluripotent stem cells all accumulate in tumors [163,164]. Radioisotope transporters, tumor lytic viruses, suicide genes, immunomodulatory agents, anti-angiogenic factors apoptosis-inducing agents, etc., are placed on the stem cells, injected locally or intravenously, and accumulate in the tumor to produce a therapeutic effect [165,166,167,168].

In suicide gene therapy, tumor cells expressing genes such as herpes simplex virus thymidine kinase (HSV-TK) and cytosine deaminase (CD) can metabolize the prodrugs, resulting in apoptosis [169]. In phase 3 clinical trial using fibroblasts with suicide genes incorporated, the significant improvement in PFS, median survival and survival rate was not observed, suggesting that the low diffusibility of suicide genes is a problem [170]. Therefore, the use of stem cells, which have a high ability to accumulate in tumors, as a vehicle for the delivery of suicide genes to glioma cells infiltrating the brain parenchyma, is being attempted for therapeutic development [165,171].

### 8.4. Viral Therapy

Antitumor therapy using oncolytic viruses (viral therapy) is a field derived from gene therapy. Proliferative viruses selectively proliferate in the tumor cells and exhibit antitumor effects through tumor lysis and inducing tumor immunity. It has been reported that the oncolytic virus has antitumor effects in mouse models of invasive tumors using GSCs due to its extensive distribution and infectivity [67]. Microenvironmental changes such as increased tumor vascular permeability and elevated expression of inflammatory cytokine genes induced by viral administration induce the resistance to viral therapy [172]. Therefore, there has been an attempt to modify the microenvironment and enhance the antitumor effects of viral therapy [143,173,174]. It is also known that tumor lytic virus activates Notch signaling in non-infected cells, and it has been reported that the combination of oncolytic virus and Notch inhibitor suppressed the cell proliferation of non-infected cells and enhanced the effect of tumor lytic virus [175]. In the future, viral therapy will continue to evolve with various studies and improvements in the approach to invasive lesions.

## 9. Conclusions

The high invasiveness of gliomas is due to a complex combination of factors, including hypoxia, ECM, cancer stem cells, EMT, etc. The impact of glioma treatment on invasiveness is not yet fully understood, and further research is needed in this area. Establishing a treatment for glioma invasion will be one of the main topics of therapeutic development in the future.

## Figures and Tables

**Figure 1 brainsci-12-00291-f001:**
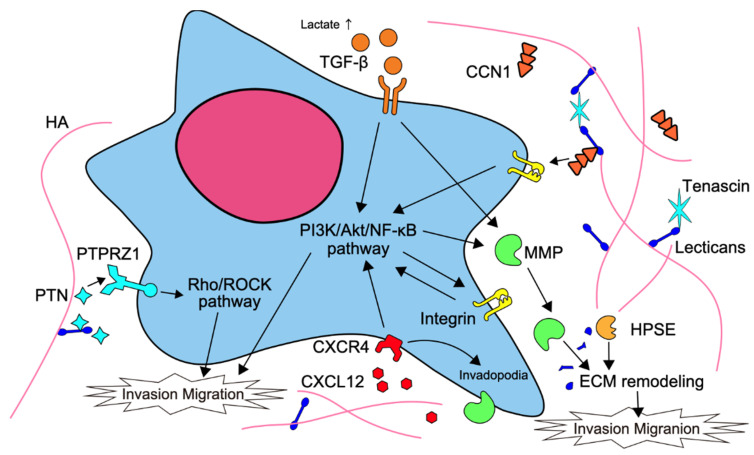
Summary of the ECM and cell surface factors involved in glioma invasion.

**Figure 2 brainsci-12-00291-f002:**
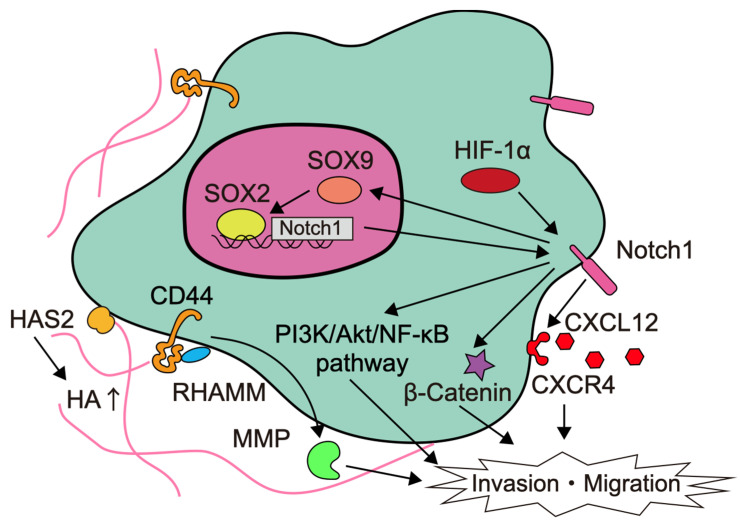
Summary of the factors involved in glioma stem cell invasion.

## Data Availability

Not applicable.

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
