# Peer review of "Molecular Mechanisms and Clinical Challenges of Glioma Invasion"

_brainsci, 2022, doi:10.3390/brainsci12020291_

Round 1

Reviewer 1 Report

  1. The authors devoted their review to the invasive properties of glioblastoma. The leading accent is made on the invasive properties of tumor cells, and insufficient attention is paid to the properties of the extracellular matrix as one of the components of the tumor microenvironment. At the same time, there are many studies that demonstrate that tumor transformation, progression and recurrence of tumors are largely determined by the characteristics of the tumor niche. More and more researchers agree with the position that in the treatment of tumor diseases it is necessary both to fight against tumor cells and create unfavorable conditions for them through antitumor resistance of the microenvironment.

  2. In the chapter 4 "Factors associated with ECM" the authors list many matrix factors, but dwell on only three of them, without focusing on chondroitin sulfate proteoglycans and hyaluronic acid as the most dominating in the matrix of the brain nervous tissue. They are shown in Figure 1. At the same time, there are collagen proteins in the Figure 1, which are localised mainly perivascularly. Collagen is traditionally considered as a component of the matrix, but the nervous tissue is fundamentally different from other tissue. The role of hyaluronic acid is discussed in the chapter 5.
  3. Moreover, the review highlights the role of PTN which being mytogenic cytokine is able to interact with glycosaminoglycans (GAGs) and is dependent on the sulfation density of GAGs.

    MMPs, considered as a «factor associated with the ECM», are activated by TGF-β, but after the shedding of GAGs from core-protein of large protein-carbohydrate molecula heparan sulfate proteoglycans by special endoglycosidase named heparanase.
  4. The authors do not proceed from the idea that the paratumorous extracellular matrix in patients with glioblastoma is modified before surgery. There are studies that demonstrate prominent changes in the properties of the ECM under the influence of dexamethasone, which affects the structure and pattern of glycosilated molecules in brain tissue. 
  5. The authors notice «Whether or not the post-radiation microenvironment enhances invasiveness is inconclusive» [218-219]. But there are results concerning at least experimental studies about changed content and pattern of proteoglycans and GAGs in irradiated brain tissue accompanied by the increase adhesion and proliferation of GBM cells.

Author Response

Response to Reviewer #1

We would like to express our appreciation to Reviewer #1 for their insightful comments, which helped us significantly improve our paper.

1)  The authors devoted their review to the invasive properties of glioblastoma. The leading accent is made on the invasive properties of tumor cells, and insufficient attention is paid to the properties of the extracellular matrix as one of the components of the tumor microenvironment. At the same time, there are many studies that demonstrate that tumor transformation, progression and recurrence of tumors are largely determined by the characteristics of the tumor niche. More and more researchers agree with the position that in the treatment of tumor diseases it is necessary both to fight against tumor cells and create unfavorable conditions for them through antitumor resistance of the microenvironment. In the chapter 4 <<Factors associated with ECM&>>; the authors list many matrix factors, but dwell on only three of them, without focusing on chondroitin sulfate proteoglycans and hyaluronic acid as the most dominating in the matrix of the brain nervous tissue. They are shown in Figure 1. At the same time, there are collagen proteins in the Figure 1, which are localised mainly perivascularly. Collagen is traditionally considered as a component of the matrix, but the nervous tissue is fundamentally different from other tissue. The role of hyaluronic acid is discussed in the chapter 5.

Response:

We are most grateful for your valuable comments on the extracellular matrix. To address this issue, we have added the following text to Chapter 4 (line 105–108). We have discussed the importance of the tumor microenvironment and added a subsection on proteoglycans (line 157–174), and have also modified Figures 1 and 2 accordingly. References have been added to reflect this change (References 58–65).

The tumor microenvironment provides gliomas with invasion, proliferation, and resistance to treatment. The extracellular matrix plays roles in scaffolding and the maintenance of tissue homeostasis, and its degradation and changes have a major impact on cancer invasion.

4.4. Proteoglycans

The major extracellular matrix components of the adult brain are glycosaminoglycan hyaluronic acid, proteoglycans of the lectican family, and link proteins [44]. Proteoglycans (PGs) are molecules consisting of a core protein and GAG side chains such as chondroitin sulfate (CS) and heparan sulfate (HS). Chondroitin sulfate proteoglycans (CSPGs) are critical regulators of brain tumor histopathology, and the low content of CSPGs is related to the active invasion of glioma cells [58]. Extracellular proteoglycans can bind to matrix proteins, trapping ligands such as growth factors [59].

The lectical subfamily of CSPG includes aggrecan, syndecan, neurocan, persican, and brevican. Syndecan-1, a transmembrane HS proteoglycan, is particularly upregulated in glioblastoma and is activated in an NFκB-dependent manner. Syndecan-1 interacts with HPSE and enhances growth factor signaling to promote the growth of glioma cells [60-62]. Moreover, brevican, a member of the lectican family of CSPG is upregulated in gliomas and its expression induces glioma invasion, which is especially enriched in the glioma stem cell (GSC) niche [63]. The HPSE is an endo-β-D-glucuronidase that degrades the heparan sulfate side chain of HSPG. It is an important regulator of ECM remodeling and is involved in the growth and invasion of glioma [64]. HPSE upregulates ERK and AKT pathways to increase glioma cell proliferation and worsen prognosis [64,65].

2) Moreover, the review highlights the role of PTN which being mytogenic cytokine is able to interact with glycosaminoglycans (GAGs) and is dependent on the sulfation density of GAGs.

Response:

Thank you for these very valuable comments. To address this, we have added the following text (line 67–71) to Chapter 2. References have also been added to reflect this change (References 24 and 25).

PTN is a strong binder of glycosaminoglycans (GAGs) and has been shown to interact with a variety of receptors, including proteoglycans PTPRZ and syndecans and GAG non-containing integrin and nucleolin [24]. These interactions depend on the sulfation density of GAGs and activate many intracellular kinases, which are involved in cell activation and transformation [24,25].

3) MMPs, considered as a «factor associated with the ECM», are activated by TGF-β, but after the shedding of GAGs from core-protein of large protein-carbohydrate molecular heparan sulfate proteoglycans by special endoglycosidase named heparanase.

Response:

We are grateful for this very useful comment. To address this, we have added the following text (line 129–133) to Chapter 4. References have also been added to reflect this change (Reference 48).

Heparanase (HPSE) degrades heparan sulfate and shortens the heparan sulfate chains on Syndecan-1, which makes the core protein susceptible to degradation by protease. Furthermore, HPSE is involved in tumor metastasis and angiogenesis by regulating the expression of downstream effector genes such as HGF, MMP-9, and VEGF [48].

4) The authors do not proceed from the idea that the paratumorous extracellular matrix in patients with glioblastoma is modified before surgery. There are studies that demonstrate prominent changes in the properties of the ECM under the influence of dexamethasone, which affects the structure and pattern of glycosilated molecules in brain tissue. 

Response:

Thank you for this very helpful comment. To address this, we have added the following text (line 327–332) to Chapter 7. A references has also been added to reflect this change (Reference 130).

DEX also affects the tumor microenvironment. The combination of TMZ and DEX affects proteoglycan structure and composition in normal brain tissue, resulting in worsened brain ECM, which is favorable for the progression of residual glioma cells; a high DEX dose results in downregulation of the transcription of PG-coding genes whereas a high DEX dose and TMZ predominantly affects the polysaccharide GAG chains of the molecules [130].

5) The authors notice «Whether or not the post-radiation microenvironment enhances invasiveness is inconclusive» [218-219]. But there are results concerning at least experimental studies about changed content and pattern of proteoglycans and GAGs in irradiated brain tissue accompanied by the increase adhesion and proliferation of GBM cells.

Response:

We appreciate this insightful comment. To address this issue, we have added the following text (line 264–268) to Chapter 7. References have also been added to reflect this change (References 102 and 103).

Radiation-induced damage of the tumor microenvironment may create tumor-susceptive niche that promotes the proliferation and invasion of the residual glioma cells [102]. In a model experiment, multiple radiations altered glycosylated components (PG and GAG) in normal brain tissue, reduced CSPG expression and CS in normal brain tissue, and promoted residual glioma cell adhesion and proliferation [103].

Reviewer 2 Report

The manuscript by Oishi et al. summarizes current knowledge on the moleccular mechanisms underlying glioma invasion. It is a clearly written and well structured paper. However, there are some concerns which may increase the value of this manuscript:

  1. It would be reasonable to explain what is the difference between te term „glioma” and „glioblastoma”; in the current version, the authors use both terms, but without such an explanation it could be misleading.
  2. The role of glutamate (its deregulated production and transport) in glioblastoma invasion (mainly due to cytotoxicity) is a well-known phenomenon. It would be reasonable to mention this issue in this manuscript.
  3. It seems that terms „invasion” and „migration” are used in paralel, which could be misleading. Please do keep in mind that some readers may not be very familiar with those terms, therefore some explanation could clarify this issue.

Author Response

Reviewer #2

The manuscript by Oishi et al. summarizes current knowledge on the molecular mechanisms underlying glioma invasion. It is a clearly written and well-structured paper. However, there are some concerns which may increase the value of this manuscript:

Response:

We would like to express our sincere appreciation to Reviewer #2 for their insightful comments, which have

helped us significantly improve our paper.

1) It would be reasonable to explain what is the difference between the term „glioma” and „glioblastoma”; in the current version, the authors use both terms, but without such an explanation it could be misleading.

Response:

Thank you for this useful comment. As per your suggestion, we have added the following text (line 23–26) to Chapter 1. A reference has also been added to reflect this change (Reference 1).

Gliomas are primary brain tumors that arise in the brain parenchyma and have histologically similar features to normal glial cells. Of these, glioblastoma is the most common tumor in adults and is a biologically aggressive tumor characterized by high cell density, pleomorphic tumors with mitosis, and either microvascular proliferation or necrosis [1].

2) The role of glutamate (its deregulated production and transport) in glioblastoma invasion (mainly due to cytotoxicity) is a well-known phenomenon. It would be reasonable to mention this issue in this manuscript.

Response:

As mentioned, glutamate is an interesting and important up-regulator of glioma invasion in the central nervous system. However, this review focuses on tumor cells and the extracellular matrix and discusses therapeutic approaches to invasion. We hope to discuss the neurotransmitter glutamate in future research.

3) It seems that terms „invasion” and „migration” are used in parallel, which could be misleading. Please do keep in mind that some readers may not be very familiar with those terms, therefore some explanation could clarify this issue.

Response:

Thank you for raising this very important point, which was not adequately explained in the original manuscript. To address this, we have added the following text (line 42–44) to Chapter 2 and modified Figure 1. A reference has also been added to reflect this change (Reference 6).

“Cell migration” is defined as the movement of cells from their original location to another whereas “cell invasion” defines the ability of cells to navigate through the extracellular matrix within a tissue or to infiltrate neighboring tissues [6].
